# A novel RSW&TST framework of MCPs detection for abnormal pattern recognition on large-scale time series and pathological signals in epilepsy

**Jinpeng Qi**[1]*, **Ying Zhu**[2], **Fang Pu**[3], **Ping Zhang**[4]

**1** College of Information Science & Technology, Donghua University, Shanghai, P.R. China, **2** HNSW Health Pathology East Genetics Level 4, Prince of Wales Hospital Randwick, Randwick, NSW, Australia, **3** Informationization Office, Donghua University, Shanghai, P.R. China, **4** Menzies Health Institute Queensland, Griffith University, Queensland, Australia

* qipengkai@dhu.edu.cn

**Data Availability Statement:** The open databses used in our research is listed as, (1) CAP Sleep Database. DOI: https://doi.org/10.13026/C2VC79 (2) Post-Ictal Heart Rate Oscillations in Partial Epilepsy. DOI: https://doi.org/10.13026/C2QC72.

## Abstract

To quickly and efficiently recognize abnormal patterns from large-scale time series and pathological signals in epilepsy, this paper presents here a preliminary RSW&TST framework for Multiple Change-Points (MCPs) detection based on the Random Slide Window (RSW) and Trigeminal Search Tree (TST) methods. To avoid the remaining local optima, the proposed framework applies a random strategy for selecting the size of each slide window from a predefined collection, in terms of data feature and experimental knowledge. For each data segment to be diagnosed in a current slide window, an optimal path towards a potential change point is detected by TST methods from the top root to leaf nodes with O (log3(N)). Then, the resulting MCPs vector is assembled by means of TST-based single CP detection on data segments within each of the slide windows. In our experiments, the RSW&TST framework was tested by using large-scale synthetic time series, and then its performance was evaluated by comparing it with existing binary search tree (BST), Kolmogorov-Smirnov (KS)-statistics, and T-test under the fixed slide window (FSW) approach, as well as the integrated method of wild binary segmentation and CUSUM test (WBS&CUSUM). The simulation results indicate that our RSW&TST is both more efficient and effective, with a higher hit rate, shorter computing time, and lower missed, error and redundancy rates. When the proposed RSW&TST framework is executed for MCPs detection on pathological ECG (electrocardiogram)/EEG (electroencephalogram) recordings of people in epileptic states, the abnormal patterns are roughly recognized in terms of the number and position of the resultant MCPs. Furthermore, the severity of epilepsy is roughly analyzed based on the strength and period of signal fluctuations among multiple change points in the stage of a sudden epileptic attack. The purpose of our RSW&TST framework is to provide an encouraging platform for abnormal pattern recognition through MCPs detection on large-scale time series quickly and efficiently.

**Funding:** This paper is supported by National Natural Science Foundation of China (No.61104154), and Specialized Research Fund for Natural Science Foundation of Shanghai (no.16ZR1401300 and no.16ZR1401200).

**Competing interests:** All authors declare that they have no known competing financial interests or personal relationships that could have appeared to influence the work reported in this paper.

## Introduction

Generally, epilepsy is a common chronic neurological disorder, and all epilepsies involve episodic abnormal electrical activity in the brain. Epilepsies, also called seizures, may be associated with cardiac arrhythmias, prominent arterial oxygen desaturations, and sudden death [1]. The authors reported that postictal heart rate oscillations are marked by the appearance of transient but prominent heart rate oscillations in a group of patients with partial epilepsy (PE). This finding may be a marker of neuroautonomic instability, and may imply some association between perturbations of the heart rate and partial seizures [1,2]. The cardiac state is generally reflected by the shape of the ECG waveform, and heart rate. Abrupt changes in heart rate can be informative and may be used as an extra clinical sign for predicting sudden epileptic attacks. In addition, sleep-related hyper-motor epilepsy (SHE), formerly known as nocturnal frontal lobe epilepsy (NFLE), is a focal epilepsy characterized by the occurrence of abrupt and typically sleep related seizures with motor patterns of variable complexity and duration [3,4]. The identification of recurrent, transient perturbations of pathological signals in brain activity during sleep, so called cyclic alternating patterns (CAP), is of significant interest as they have been linked to multiple pathologies [3,5].

Pathological signals can be recorded and processed by using present signal processing techniques, such as time series analysis [6], fast Fourier transform (FFT) [7], power spectral density (PSD) [8], empirical mode decomposition (EMD) [9,10] wavelet analysis [11] etc. However, the enormous volume of data usually makes the study tedious and time-consuming for these traditional methodologies. Abnormal patterns or change points in the signals can indicate that important events have occurred, or that a system has changed in critical ways [12,13]. Change point detection has been widely studied and has been applied to help medical research, for example, through the prediction of onset of illness or increasing illness severity, gene expression detection, and to other fields (eg. climate science) [14–20].

Sliding window strategies are very useful tools for multiple change points (MCPs) detection in signal processing, and have been investigated in various fields [21–26]. However, a key factor of these strategies is how to select a suitable size of sliding window, because it needs to capture the necessary characteristics of a signal to achieve correct detection/classification [24]. If the window size is too small, the task of pattern recognition will be split into multiple consecutive windows without achieving high efficiency. On the other hand, if the window size is too large, it might contain multiple patterns and decrease the recognition performance [22,24]. In addition, wild binary segmentation (WBS) method is a popular technique for multiple change-point detection [27,28]. By using the CUSUM-like test in the stochastic manner, WBS avoid the problem of span or window selection by drawing intervals of different lengths [27,28]. However, WBS might encounter a problem that the regular and rhythmic data fluctuations in the entire datasets might be redundantly selected, even the actual target change points (tCPs) might be missed or discarded, especially when the criterion of candidate change point is unsuitable.

In this paper a novel RSW&TST framework for MCPs detection is presented based on random slide window (RSW) and trigeminal search tree (TST) methods [18,20,29]. With the RSW approach, a series of slide windows in random sizes is selected according to the data features and experimental knowledge. The TST method is used to detect a series of single change points from each of the slide windows and to create a vector of MCPs. Based on synthetic time series, our RSW&TST method was evaluated by comparing it to existing FSW and WBS&CUSUM, as well as BST, KS and T methods [15,18,20,23,30], in terms of computing time, and the values of hit, missed, error, and redundancy, rates. In real experiments on pathological recordings in epilepsy, our RSW&TST was applied to not only recognize the abnormal patterns in

terms of the number and position of the resultant MCPs, but also roughly estimate the severity of the epilepsy in accordance with the strength and period of signal fluctuations among MCPs vector during a sudden epileptic attack.

## Backgrounds

### Definition of MCPs

Suppose a time-series signal $X = (X_1,\ldots,X_i,\ldots,X_N)$ can be observed as a trajectory of a multiple data distribution process, in which the segment $X_i$ is defined by [31–33]:

$$X_i = f_i(t) + \varepsilon_i, \tag{1}$$

where $t \in \{t_{i-1}+1,\ldots, t_i\}$, $0 < i <= M$, and $f_i \in \{f_1,\ldots,f_M\}$ is a deterministic and piece-wise function of one-dimensional signals with MCPs (satisfying $f_i \neq f_{i+1}$, and $i = 1,\ldots, M$-1 for insuring that changes occur), and $M \in \{1, 2,\ldots, n\}$ is the number of data segment regimes and therefore $M$-1 is the number of abrupt changes, $0 = t_0 < t_1 < \cdots < t_i < \cdots < t_M = n$. The number $M$-1 and locations $\eta_1,\ldots, \eta_{M-1}$ of MCPs in the process are supposed to be unknown. The sequence $(\varepsilon_i)_{i \in N}$ is assumed to be random white noise and such that $E(\varepsilon_i)$ is exactly or approximately zero. In the simplest case $(\varepsilon_i)_{i \in N}$ is modeled as i.i.d., but can also follow more complex time series distributions [33].

### Wild binary segmentation (WBS)

Initially, the wild binary segmentation (WBS) method randomly draws a number of vectors $(X_s, X_{s+1},\ldots,X_e)$ from the entire data sample $(X_1, X_2,\ldots,X_T)$, where $s$ and $e$ are integers such that $1 \leq s < e \leq T$, and then compute the CUSUM-like test on each subsample [27]. The whole dataset is split into two sub segments, if and once a change-point is detected typically via CUSUM-like procedure [27]. Then, the WBS choose the largest one over the entire collection of statistical tests, and take it to be the first change-point candidate to be tested against a certain threshold. If it is considered to be significant, the same procedure is then repeated recursively to the left and to the right.

By using the CUSUM test in the stochastic manner, WBS avoids the problem of span or window selection by drawing intervals of different lengths [27,28]. However, WBS probably encounters an issue as the entire dataset contains the regular and rhythmic data fluctuations. It seems reasonable that the largest maximiser from the entire collection of CUSUM-like tests is taken to be the first change-point candidate against a certain threshold, but the rest part satisfied with the candidate criterion might be redundantly selected, even the acutal target change points (tCPs) might be missed or discarded in the process of MCPs detection, especially when the threshold for candidate change point in the collection of CUSUM test is too low or too high.

### TST-based single CP detection

In the RSW approach, as shown in Fig 1, we consider a diagnosed time series signal $X' = (X_s,\ldots,X_i,\ldots,X_e)$ is divided into multiple data segments accordingly by random slide windows. As for an observed time series segment $X^i = \{X_a^i, \ldots, X_c^i, \ldots, X_b^i\}$ in each slide window $W_i$, we apply the TST method for a potential change point detection from $W_i$ in our RSW framework [22,29]. In this TST-base single CP detection method, as shown in Fig 2, the trigeminal search trees TSTcA/TSTcD are first constructed by adding virtual middle branches into existing binary trees [18,20]. Then, the search criteria for multi-channel detection are executed to find an optimal path towards a potential change point from the top root to the bottom leaf

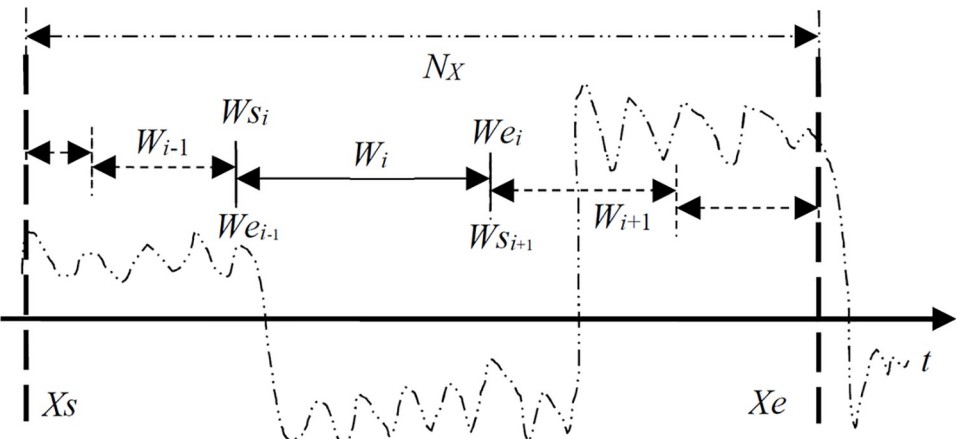

**Fig 1. The scheme of the RSW approach for MCPs detection on time series sample X.**

levels in the TSTcA/TSTcD respectively. Finally, a resultant change point is obtained from $X^i$ in the current $W_i$ after $\log Nw_i$ search steps, where $Nw_i$ is the length of $X^i$.

**TSTs construction.** If the length $Nw_i$ of $X^i$ is $k$ times divisible by 2, then $X^i$ can be generally decomposed into an average signal vector $A^k$ and a set of detail signal vectors $D = \{D^1, D^2,\ldots,D^k\}$, and then represented by a mapping $H_k$ in terms of the $k$-level Haar Wavelet Transform (HWT) as follows:

$$X^i \xrightarrow{H_k} (A^k|D^k|D^{k-1}|\ldots|D^2|D^1), \tag{2}$$

where $1 \leq k \leq Lk = \log_2 Nw_i$, and $Nw_i = |b-a| = |We_i-Ws_i|$. In terms of Multi-Resolution Analysis (MRA) [34,35], in HWT, the average vector $A^k$ and the set of detail vectors $D = \{D^k, D^{k-1},\ldots,D^2, D^1\}$ can be conceptualized as a projection of time series $X$ to the different scaling vectors $V^k$ and a series of wavelet basis vectors $W = \{W^k, W^{k-1},\ldots,W^2, W^1\}$. The vectors $A^k$ and

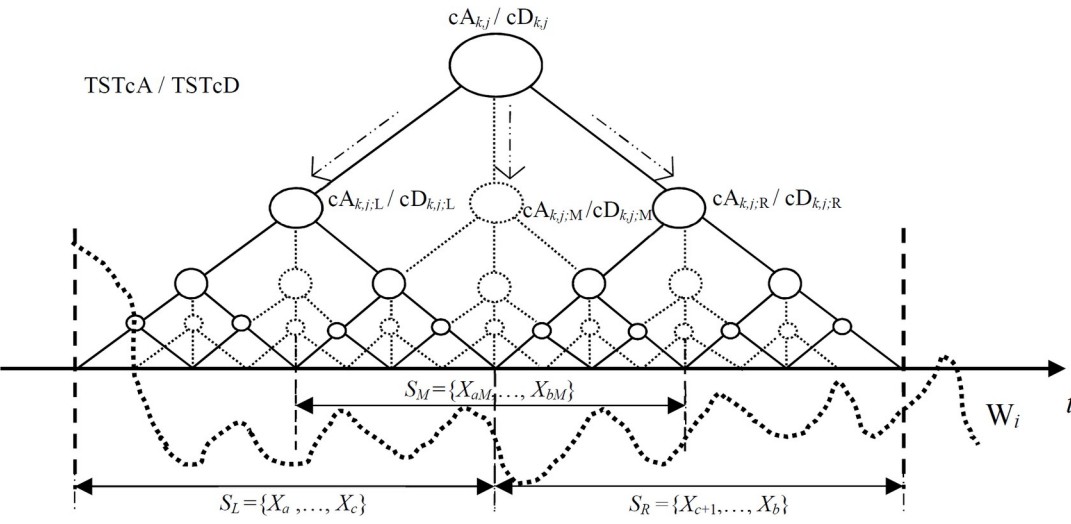

**Fig 2. The scheme of TSTcA/TSTcD construction by adding the virtual middle branches cA_{k,j:M}/cD_{k,j:M} into the existing binary trees in each of non-leaf levels.**

$D^k$ can be expressed as follows:

$$A^k = (X \cdot V^k)V^k = \sum_{j=1}^{Lj}(X \cdot v_j^k)v_j^k = \sum_{j=1}^{Lj}(cA_{k,j})v_j^k, \tag{3}$$

$$D^k = (X \cdot W^k)W^k = \sum_{j=1}^{Lj}(X \cdot w_j^k)w_j^k = \sum_{j=1}^{Lj}(cD_{k,j})w_j^k, \tag{4}$$

where $v_j^k$ is the $j^{\text{th}}$ level signal of scaling vector $V^k$, and $w_j^k$ is the $j^{\text{th}}$ level signal of wavelet basis vector $W^k$; $|v_j^k| = |w_j^k| = N$, $Lj = N/2^k$, and $k = 1,2,\ldots,\log_2 Nw_i$. The coefficient vectors in the average signal set $A = \{A^k | 0 \leq k \leq Lk\}$ and the detail signal set $D = \{D^k | 0 \leq k \leq Lk\}$ can be further presented by the following two matrices McA and McD:

$$\text{McA} = \begin{bmatrix} A^0 \\ A^1 \\ \cdots \\ A^k \\ \cdots \\ A^{Lk} \end{bmatrix} = \begin{bmatrix} cA_{0,1} & \cdots & \cdots & \cdots & \cdots & cA_{0,N} \\ cA_{1,1} & \cdots & \cdots & \cdots & cA_{1,N/2} & \\ \cdots & \cdots & \cdots & \cdots & & \\ cA_{k,1} & \cdots & cA_{k,m} & & & \\ \cdots & \cdots & & & & \\ cA_{Lk,1} & & & & & \end{bmatrix}, \tag{5}$$

$$\text{McD} = \begin{bmatrix} D^0 \\ D^1 \\ \cdots \\ D^k \\ \cdots \\ D^{Lk} \end{bmatrix} = \begin{bmatrix} cD_{0,1} & \cdots & \cdots & \cdots & \cdots & cD_{0,N} \\ cD_{1,1} & \cdots & \cdots & \cdots & cD_{1,N/2} & \\ \cdots & \cdots & \cdots & \cdots & & \\ cD_{k,1} & \cdots & cD_{k,n} & & & \\ \cdots & \cdots & & & & \\ cD_{Lk,1} & & & & & \end{bmatrix}, \tag{6}$$

where $1 \leq j \leq Lj$, $1 \leq k \leq Lk$, and $A^0 = D^0 = X^i = \{X_a^i, \ldots, X_c^i, \ldots, X_b^i\}$. Based on both McA and McD, as shown in Fig 2, the TSTcA and TSTcD are constructed by adding the virtual middle branches at each non-leaf level in the existing TcA and TcD. Therefore, the time series segment $X^i$ can be divided into three overlapped parts of $S_L = \{X_a^i, \ldots, X_c^i\}$, $S_M = \{X_{aM}^i, \ldots, X_{bM}^i\}$, and $S_R = \{X_{c+1}^i, \ldots, X_b^i\}$.

If a current non-leaf node $cA_{k,j}/cD_{k,j}$ is selected in both TSTcA/TSTcD, the related variables are denoted by the following formulas:

$$cA_{k,j:L} = cA_{k-1,2j-1} = \frac{1}{(\sqrt{2})}\left(cA_{k-2,4j-3} + cA_{k-2,4j-2}\right), \tag{7}$$

$$cA_{k,j:R} = cA_{k-1,2j} = \frac{1}{(\sqrt{2})}\left(cA_{k-2,4j-1} + cA_{k-2,4j}\right), \tag{8}$$

$$cA_{k,j:M} = \frac{1}{(\sqrt{2})}\left(cA_{k-2,4j-2} + cA_{k-2,4j-1}\right), \tag{9}$$

$$cD_{k,j:L} = cD_{k-1,2j-1} = cD_{k-1,(4j-3;4j-2)} = \frac{1}{(\sqrt{2})}\left(cA_{k-2,4j-3} - cA_{k-1,4j-2}\right), \tag{10}$$

$$cD_{k,j:R} = cD_{k-1,2j} = cD_{k-1,(4j-1;4j)} = \frac{1}{(\sqrt{2})}\left(cA_{k-2,4j-1} - cA_{k-1,4j}\right), \tag{11}$$

$$cD_{k,j:M} = \frac{1}{(\sqrt{2})}\left(cA_{k-2,4j-2} - cA_{k-1,4j-1}\right), \tag{12}$$

$$cA_{k,j} = \frac{1}{(\sqrt{2})^k}\left(\sum\nolimits_{n=a}^{b} X_n^i\right), \tag{13}$$

$$cD_{k,j} = \frac{1}{(\sqrt{2})^k}\left(\left(\sum\nolimits_{l=a}^{c} X_l^i - \sum\nolimits_{r=c+1}^{b} X_r^i\right)\right)$$

$$= \frac{1}{(\sqrt{2})}\left(\frac{1}{(\sqrt{2})^{k-1}}\left(\sum\nolimits_{l=a}^{c} X_l^i - \sum\nolimits_{r=c+1}^{b} X_r^i\right)\right)$$

$$= \frac{1}{(\sqrt{2})}\left(cA_{k-1,2j-1} - cA_{k-1,2j}\right), \tag{14}$$

where $2 \leq k \leq Lk$ and $1 \leq j \leq Lj$; $a = 2^k(j-1)+1$, $b = 2^{k*}j$, and $c = 2^k(j-1)+2^{(k-1)}$. The implementation of the TSTcA/TSTcD construction is described in Algorithm 1 in detail.

**Trigeminal-branches search strategies.** To detect a potential change point from a time series segment in each of the slide windows quickly and efficiently, three search criteria are introduced on the basis of data features within existing TSTcA/TSTcD at different non-leaf levels.

*TSTcD-based search criterion.* **Definition 2.1:** Suppose the time series segment $X^i = \{X_a^i, \ldots, X_c^i, \ldots, X_b^i\}$ in a current slide window $W_i$ is selected from the whole observed sample $X = \{X_1, \ldots, X_N\}$. Then the variance fluctuation (VF) within $X^i$ is defined by:

$$VF_{mn}(X_c^i) = \sup_{c \in \mathbb{Z}, a < c < b} \left|\frac{1}{m}\sum\nolimits_{l=a}^{c} X_l^i - \frac{1}{n}\sum\nolimits_{r=c+1}^{b} X_r^i\right|, \tag{15}$$

where $1 \leq a < b \leq N$, $a \leq l \leq c$, $c+1 \leq r \leq b$, $m = c-a+1$, and $n = b-c$. If $VF_{mn}(X_c^i) > C_1(\alpha)$ holds, then an abrupt change occurs at the time point $X_c^i$ between two adjacent data segments $X_L^i = \{X_a^i, \ldots, X_c^i\}$ and $X_R^i = \{X_{c+1}^i, \ldots, X_b^i\}$ in $X^i$, and $C_1(\alpha) \in R$ represents a threshold of the variance fluctuation between $X_L^i$ and $X_R^i$ which obey an identical distribution in $X^i$. On the other hand, if $VF_{mn}(X_c^i) \leq C_1(\alpha)$ holds, then no abrupt change occurs in the current segment $X^i$.

**Definition 2.2:** Given a piece of a time series segment $X^i$ in the slide window $W_i$, suppose a sub-tree $cD_{k,j}$ is selected from the trigeminal search tree TSTcD at non-leaf node levels, three variance fluctuations $VF_{k,j:L}$, $VF_{k,j:R}$, and $VF_{k,j:M}$ are formulated in accordance with the sub-

branches $cD_{k,j:L}$, $cD_{k,j:R}$ and $cD_{k,j:M}$ as follows:

$$VF_{k,j:L} = \left| \frac{1}{2^{k-2}} \left( \sum_{l=a}^{aM} X_l^i - \sum_{r=aM+1}^{c} X_r^i \right) \right|$$

$$= \left(\frac{1}{\sqrt{2}}\right)^{k-3} \left| \frac{1}{(\sqrt{2})^{k-1}} \left( \sum_{l=a}^{aM} X_l^i - \sum_{r=aM+1}^{c} X_r^i \right) \right|$$

$$= \left(\frac{1}{\sqrt{2}}\right)^{k-3} |cD_{k-1,2j-1}| = \left(\frac{1}{\sqrt{2}}\right)^{k-3} |cD_{k,j:L}|, \tag{16}$$

$$VF_{k,j:R} = \left| \frac{1}{2^{k-2}} \left( \sum_{l=c+1}^{bM} X_l^i - \sum_{r=bM+1}^{b} X_r^i \right) \right|$$

$$= \left(\frac{1}{\sqrt{2}}\right)^{k-3} \left| \frac{1}{(\sqrt{2})^{k-1}} \left( \sum_{l=c+1}^{bM} X_l^i - \sum_{r=bM+1}^{b} X_r^i \right) \right|$$

$$= \left(\frac{1}{\sqrt{2}}\right)^{k-3} |cD_{k-1,2j}| = \left(\frac{1}{\sqrt{2}}\right)^{k-3} |cD_{k,j:R}|, \tag{17}$$

$$VF_{k,j:M} = \left| \frac{1}{2^{k-2}} \left( \sum_{l=aM+1}^{c} X_l^i - \sum_{r=c+1}^{bM} X_r^i \right) \right|$$

$$= \left(\frac{1}{\sqrt{2}}\right)^{k-3} \left| \frac{1}{(\sqrt{2})^{k-1}} \left( \sum_{l=aM+1}^{c} X_l^i - \sum_{r=c+1}^{bM} X_r^i \right) \right|$$

$$= \left(\frac{1}{\sqrt{2}}\right)^{k-3} |cD_{k,j:M}|, \tag{18}$$

where $1 \leq j \leq N/2^k$, $m = n = 2^{k-2}$, and $2 \leq k \leq log_2 N$; $a = 2^k(j-1)+1$, $b = 2^k*j$, and $c = 2^k(j-1)+2^{k-1}$; $aM = 2^{k-1}(2j-2)+2^{k-2}$, $bM = 2^{k-1}(2j-1)+2^{k-2}$.

**Criterion 2.1:** Given three measurements $VF_{k,j:L}$, $VF_{k,j:R}$, and $VF_{k,j:M}$ in definition 2.2, if $max(VF_{k,j:L}, VF_{k,j:R}, VF_{k,j:M}) > C_1(\alpha)$ and $2 \leq k \leq log_2 N$ hold, then the sub-branch with the maximal VF value is selected from $cD_{k,j:L}$, $cD_{k,j:R}$, or $cD_{k,j:M}$ in the TSTcD, and two others are discarded.

**Proof 2.1:** Suppose a target CP $X_c^i$ is contained in an observed segment $X^i = \{X_a^i, \ldots, X_c^i, \ldots, X_b^i\}$. Then, in terms of Definition 2.1, there exists a bigger VF value between two adjacent segments before and after the target $X_c^i$, than that of any other parts without $X_c^i$. As for a current non-leaf node $cD_{k,j}$ with trigeminal branches $cD_{k,j:L}$, $cD_{k,j:R}$ and $cD_{k,j:M}$ in the TSTcD, in terms of the definition 2.2, one reliable explanation for Criterion 2.1 can be associated with the reason that the one with the maximal VF value in all sub-branches $cD_{k,j:L}$, $cD_{k,j:R}$ and $cD_{k,j:M}$ has a higher probability of containing the target CP than that of two others. Therefore, it is reasonable to select the sub-branch with the maximal VF value as the current search path and discard the others.

*TSTcA-based search criterion.* **Definition 2.3:** Suppose a piece of data segment $X^i = \{X_a^i, \ldots, X_c^i, \ldots, X_b^i\}$ in a current slide window $W_i$ is selected to be diagnosed from the whole time series sample $X = \{X_1, \ldots, X_N\}$, then the statistic fluctuation (SF) is defined as follows:

$$SF_{mn}(X_c^i) = \sup_{c \in \mathbb{Z}, a < c < b} (\frac{mn}{m+n})^{\frac{1}{2}} |F_m(X_c^i) - G_n(X_c^i)|$$

$$= \sup_{c \in \mathbb{Z}, a < c < b} (\frac{mn}{m+n})^{\frac{1}{2}} |\frac{1}{m}\sum_{l=a}^c I(X_l^i \leq X_c^i)\} - \frac{1}{n}\sum_{r=c+1}^b I(X_r^i \leq X_c^i)\}|, \quad (19)$$

where $1 \leq a < b \leq N$, $a \leq l \leq c$, $c+1 \leq r \leq b$, $m = c-a+1$, and $n = b-c$. $F_m$ and $G_n$ stand for the empirical cumulative distribution function (e.c.d.f) of two adjacent data segments $X_L^i = \{X_a^i, \ldots, X_c^i\}$ and $X_R^i = \{X_{c+1}^i, \ldots, X_b^i\}$ respectively, and $I$ is an indicator function. Suppose $SF_{mn}(X_c^i) > C_2(\beta)$ holds, there exists an abrupt change $X_c^i$ within $X^i$, where $C_2(\beta) \in R$ is a threshold of the SF value between $X_L^i$ and $X_R^i$ that obey an identical distribution. On the other hand, if $SF_{mn}(X_c^i) \leq C_2(\beta)$ holds, then no abrupt change exists in the current segment $X^i$.

**Definition 2.4:** Consider the other TSTcA constructed from the same data segment $X^i$ in $W_i$. Suppose a sub-tree $cA_{k,j}$ is selected from one of the non-leaf nodes in TSTcA, then the related variables $X_{k,j:L}^i, X_{k,j:R}^i$ and $X_{k,j:M}^i$ are introduced, and three statistic fluctuations $SF_{k,j:L}, SF_{k,j:R}$, and $SF_{k,j:M}$ are presented according to the three sub-branches $cA_{k,j:L}, cA_{k,j:R}$ and $cA_{k,j:M}$ as follows:

$$X_{k,j:L}^i = \frac{1}{2^{k-1}}\left(\sum_{l=a}^c X_l^i\right) = (\frac{1}{\sqrt{2}})^{k-1} cA_{k,j:L} = (\frac{1}{\sqrt{2}})^{k-1} cA_{k-1,2j-1}, \quad (20)$$

$$X_{k,j:R}^i = \frac{1}{2^{k-1}}\left(\sum_{r=c+1}^b X_r^i\right) = (\frac{1}{\sqrt{2}})^{k-1} cA_{k,j:R} = (\frac{1}{\sqrt{2}})^{k-1} cA_{k-1,2j}, \quad (21)$$

$$X_{k,j:M}^i = \frac{1}{2^{k-1}}\left(\sum_{m=aM+1}^{bM} X_m^i\right) = (\frac{1}{\sqrt{2}})^{k-1} cA_{k,j:M} = (\frac{1}{\sqrt{2}})^k (cA_{k-2,4j-2} + cA_{k-2,4j-1}), \quad (22)$$

$$SF_{k,j:L} = SF_{mn}\left(X_{k,j:L}^i\right) = (\frac{mn}{m+n})^{\frac{1}{2}} |\{\frac{1}{m}\sum_{l=a}^{aM} I(X_l^i \leq X_{k,j:L}^i) - \frac{1}{n}\sum_{r=aM+1}^c I(X_r^i \leq X_{k,j:L}^i)\}|, \quad (23)$$

$$SF_{k,j:R} = SF_{mn}\left(X_{k,j:R}^i\right)$$
$$= (\frac{mn}{m+n})^{\frac{1}{2}} |\{\frac{1}{m}\sum_{l=c+1}^{bM} I(X_l^i \leq X_{k,j:R}^i) - \frac{1}{n}\sum_{r=bM+1}^b I(X_r^i \leq X_{k,j:R}^i)\}|, \quad (24)$$

$$SF_{k,j:M} = SF_{mn}\left(X_{k,j:M}^i\right)$$
$$= (\frac{mn}{m+n})^{\frac{1}{2}} |\{\frac{1}{m}\sum_{l=aM+1}^c I(X_l^i \leq X_{k,j:M}^i) - \frac{1}{n}\sum_{r=c+1}^{bM} I(X_r^i \leq X_{k,j:M}^i)\}|, \quad (25)$$

where $a = 2^k(j-1)+1$, $b = 2^k{}_*j$, and $c = 2^k(j-1)+2^{k-1}$; $aM = 2^{k-1}(2j-2)+2^{k-2}$, $bM = 2^{k-1}(2j-1)+2^{k-2}$, $m = n = 2^{k-2}$, $1 \leq j \leq N/2^k$, and $2 \leq k \leq \log_2 N$.

**Criterion 2.2:** Consider the three variables $SF_{k,j:L}, SF_{k,j:R}$, and $SF_{k,j:M}$ in Definition 2.3. If $max(SF_{k,j:L}, SF_{k,j:R}, SF_{k,j:M}) > C_2(\alpha)$ and $2 \leq k \leq \log_2 N$ hold, then the sub-branch with the maximal SF value is selected from $cA_{k,j:L}, cA_{k,j:R}$, and $cA_{k,j:M}$ in the TSTcA, and the others are omitted.

**Proof 2.2:** Suppose a change point $X_c^i$ exists in an observed segment $X^i = \{X_a^i, \ldots, X_c^i, \ldots, X_b^i\}$. Then, in terms of Definition 2.3, the SF value between two adjacent segments containing $X_c^i$ will be bigger than that of any part without $X_c^i$. On the other hand, according to Definition 2.4, Criterion 2.2 reliably shows that the part with the maximal SF value in all sub-branches $cA_{k,j:L}$, $cA_{k,j:R}$, and $cA_{k,j:M}$ has a higher probability of including the target CP than the other parts. As a result, it is best to choose the sub-branch with the maximal SF value as the current search path, and dismiss the others.

*Leaf-node search criterion.* To find a target CP from the bottom leaf nodes in the TSTcA/TSTcD, another search criterion is introduced by using the revised KS statistics.

**Definition 2.5:** Given a sub-tree $cA_{k,j}/cD_{k,j}$ selected from TSTcA/TSTcD in the last non-leaf level, where $k = 1$, two statistic variables $S_L(X_{cL}^i)$ and $S_R(X_{cR}^i)$ are defined in terms of two leaf nodes $cA_{0,2j-1}/cD_{0,2j-1}$ and $cA_{0,2j}/cD_{0,2j}$ as follows:

$$S_L(X_{cL}^i) = S_{mn}(X_{cL}^i) = (\frac{mn}{m+n})^{\frac{1}{2}}|F_m(X_{cL}^i) - G_n(X_{cL}^i)|$$

$$= (\frac{mn}{m+n})^{\frac{1}{2}}|\{\frac{1}{m}\sum_{l=a}^{2j-1} I(X_l^i \leq X_{cL}^i) - \frac{1}{n}\sum_{r=2j}^{b} I(X_r^i \leq X_{cL}^i)\}|, \tag{26}$$

$$S_R(X_{cR}^i) = S_{mn}(X_{cR}^i) = (\frac{mn}{m+n})^{\frac{1}{2}}|F_m(X_{cR}^i) - G_n(X_{cR}^i)|$$

$$= (\frac{mn}{m+n})^{\frac{1}{2}}|\{\frac{1}{m}\sum_{l=a}^{2j} I(X_l^i \leq X_{cR}^i) - \frac{1}{n}\sum_{r=2j+1}^{b} I(X_r^i \leq X_{cR}^i)\}|, \tag{27}$$

where $X_{cL}^i = cA_{0,2j-1} = cD_{0,2j-1} = X_{2j-1}^i$, and $X_{cR}^i = cA_{0,2j} = cD_{0,2j} = X_{2j}^i$; $F_m(X_c^i)$ and $G_n(X_c^i)$ refer to the e.c.d.f of two data segments $X_L^i = \{X_a^i, \ldots, X_c^i\}$ and $X_R^i = \{X_{c+1}^i, \ldots, X_b^i\}$ respectively; $cL = 2j-1$, $cR = 2j$, $m = 2j-1$ or $2j$, and $n = N-m+1$.

In addition, it is worth noting that the largest SF between $F_m(x)$ and $G_n(x)$ is achieved either before or after one of the signal jumps, i.e., an abrupt change, as:

$$S_{mn}(x) = \sup_{x \in R}|G_n(x) - F_m(x)| = \max_{1 \leq i \leq N} \begin{cases} |F_m(X_i^-) - G_n(X_i^-)|, & \text{before the } i\text{th jump} \\ |F_m(X_i) - G_n(X_i)|, & \text{after the } i\text{th jump} \end{cases}, \tag{28}$$

**Definition 2.6:** In terms of the formula (34) in Definition 2.5, we then define another two variables $S_L^-$ and $S_R^-$ as,

$$S_L^-(X_{cL}^i) = S_{mn}(X_{cL}^{i-}) = (\frac{mn}{m+n})^{1/2}|F_m(X_{cL}^{i-}) - G_n(X_{cL}^{i-})|$$

$$= (\frac{mn}{m+n})^{\frac{1}{2}}|\{\frac{1}{m}\sum_{l=a}^{2j-1} I(X_l^i < X_{cL}^i) - \frac{1}{n}\sum_{r=2j}^{b} I(X_r^i < X_{cL}^i)\}| \tag{29}$$

$$S_R^-(X_{cR}^i) = S_{mn}(X_{cR}^{i-}) = \left(\frac{mn}{m+n}\right)^{1/2}|F_m(X_{cR}^{i-}) - G_n(X_{cR}^{i-})|$$

$$= (\frac{mn}{m+n})^{\frac{1}{2}}|\{\frac{1}{m}\sum_{l=a}^{2j} I(X_l^i < X_{cR}^i) - \frac{1}{n}\sum_{r=2j+1}^{b} I(X_r^i < X_{cR}^i)\}| \tag{30}$$

Thereafter, the maximal values of two statistic measurements in accordance with the leaf nodes $cA_{0,2j-1}/cD_{0,2j-1}$ and $cA_{0,2j}/cD_{0,2j}$ can be obtained by $S'_L = max(S^-_L, S_L)$ and $S'_R = max(S^-_R, S_R)$, respectively.

**Criterion 2.3:** For two statistic variables $S'_L$ and $S'_R$ above, if $max(S'_L, S'_R) > C_3(r)$ holds, then one of the two leaf nodes $cA_{0,2j-1}/cD_{0,2j-1}$ and $cA_{0,2j}/cD_{0,2j}$ with $max(S'_L, S'_R)$ is chosen from TSTcA/TSTcD at the last bottom level, and the other one is discarded. Accordingly, one of two time points $X^i_{2j-1}$ or $X^i_{2j}$ is selected from the diagnosed segment $X^i$, and dealt as the final resultant CP. Otherwise, no abrupt change is detected from the current slide window $W_i$.

**Proof 3.3:** Providing a change point $X^i_c$ exists in an observed segment $X^i = \{X^i_a, \ldots, X^i_c, \ldots, X^i_b\}$, the values of $S'_L$ and $S'_R$ can be calculated precisely according to Definition 2.5 and 2.6. Criterion 2.3 ensure that the leaf node with $max(S'_L, S'_R)$ is selected as the resultant CP, because it has a higher probability of containing the target CP than the other one. Meanwhile, if $max(S'_L, S'_R) > C_3(r)$ holds, then the statistic distance between two adjacent parts $X^i_L = \{X^i_a, \ldots, X^i_c\}$ and $X^i_R = \{X^i_{c+1}, \ldots, X^i_b\}$ has exceeded the threshold value $C_3(r)$ that $X^i_L$ and $X^i_R$ belong to an identical distribution. Therefore, it can be guaranteed that one of two time points $X^i_{2j-1}$ or $X^i_{2j}$ with $max(S'_L, S'_R)$ is chosen from the diagnosed time series segment $X^i$, and then dealt as the final resultant CP detected from an observed slide window $W_i$, and the other one is ignored.

## Proposed method

### RSW&TST framework

In the proposed RSW&TST framework, the RSW approach is applied for dividing the diagnosed time series $X' = (X_s, \ldots, X_i, \ldots, X_e)$ mentioned above randomly into multiple data segments. Then, the TST-base method is executed repeatedly to detect the potential single CP from each slide window. Our RSW&TST framework for MCPs detection is stated as below.

1. First, given a slide window $W_{i-1}$ shown in Fig 1, the data segment is denoted as $X^{i-1} = [X^{i-1}_{Ws_{i-1}}, \ldots, X^{i-1}_{We_{i-1}}]$, and the total sample length $TN\_w_{i-1}$ from the beginning slide window $W_1$ to the current one $W_{i-1}$ is denoted as,

$$0 < TN\_w_{i-1} = \sum_{k=1}^{i-1}(Nw_k) \leq N_{X'}, \tag{31}$$

   where $Nw_k = |We_k - Ws_k|$, $1 \leq k \leq i-1$, and $N_{X'} = length(X')$.

2. As for a successive slide window $W_i$, the candidate set of slide window size $Set\_Nw_i$ is defined as below,

$$Set\_Nw_i = \{Nw_i | Cd <= Nw_i <= Cu\}, \tag{32}$$

   where $Cd = Td\_Nw_i$ and $Cu = Tu\_Nw_i$ are two predefined constants, and $0 < Cd \leq Cu < N_{X'}$, referring to the lower and upper bounds of $Nw_i$ respectively, and $1 < i < n$, $n$ is the number of data segments within $X'$.

3. Next, the rest length of unprocessed part $N_R$ from the beginning of $W_i$ to the end of $X'$ is presented by,

$$N_R = N_{X'} - TN\_w_{i-1}, \tag{33}$$

   where $TN\_w_{i-1}$ and $N_{X'}$ are defined as in step (1).

4. Then, the candidate set $Set\_Nw_i$ can be reformulated as,

$$Set\_Nw_i = \{Nw_i | Cd <= Nw_i <= Cu \leq N_R\}, \text{ if } N_R \geq Cu, \tag{34}$$

$$Set\_Nw_i = \{Nw_i | Cd <= Nw_i <= N_R < Cu\}, \text{ if } Cd \leq N_R < Cu, \tag{35}$$

In addition, if $0 < N_R < Cd$ holds, then the process of MCPs detection jumps to step (8), and the RSW approach is ended.

5. Next, the size of slide window $Nw_i$ can be selected randomly from $Set\_Nw_i$ mentioned above. This step is denoted by,

$$Nw_i = random(Set\_Nw_i), \tag{36}$$

where $random(SetNw_i)$ is a pseudo-function to select a random value of $Nw_i$ from $Set\_Nw_i$, and the two endpoints of slide window $W_i$ are readjusted by $Ws_i = We_{i-1}+1$, and $We_i = Ws_i+Nw_i$, respectively.

6. Thereafter, the data segment within current slide window $W_i$, $X^i = [X^i_{Ws_i}, \ldots, X^i_{We_i}]$, is disposed of by the TST-base approach for a potential CP detection. If there exists a change point $CP_i$ in $W_i$, then the $CP_i$ is assembled into the resultant MCPs vector in order.

7. Similarly, the procedure of TST-based single CP detection is repetitively executed on a series of data segments in the successive slide windows, until $0 < N_R < Cd$ holds. That is, the TST-based approach for CP detection is stopped when the rest part in the last slide window is less than the lower bound of the minimal window size $Cd$.

8. Finally, the vector of resultant MCPs is assembled by all the detected CPs from slide windows, and then the RSW&TST framework for MCPs detection is coming to an end.

## Implementations of RSW&TST framework

In the implementations of our RSW&TST framework, the proposed RSW approach aims to divide the whole time series into a series of data segments by random slide windows. For each slide window, the procedure of TST-based single CP detection is executed for discerning an optimal path towards a potential change point from the TSTcD/TSTcA. In this multi-channel search process, Criterion 2.1 is used for selecting the abnormal part from trigeminal branches in the TSTcD at each non-leaf level. Criterion 2.2 is used for discerning the abnormal one from trigeminal branches in the TSTcA at each non-leaf level, if Criterion 2.1 is invalid as the value of VF measurement is indistinctive to be detected. Criterion 2.3 is executed to estimate a potential CP from the left and right nodes in the last leaf level. Finally, the vector of resultant MCPs is assembled orderly by a series of detected CPs from each of slide windows. The related algorithms including the procedures of TSTs construction, and TST-based single CP detection, as well as the integrated framework of RSW&TST for MCP detection are described in Algorithms 1–3 in detail.

```
Algorithm 1: TSTs construction.
Input: a piece of time series segment X^i = {X^i_a,...,X^i_c,...,X^i_b} within an
observed slide window W_i.
Initialization: N = length(X^i), K = log_2 N, J = N/2^K;
j = 1, k = 1;
Declare matrices McA MVcA, McD, and MVcD;
For i = 1 to k do
    [cA_i, cD_i] = Call Haarwavelet(X, i);
    McA(i) = cA_i; McD(i) = cD_i
End for
For i = 2 to k do
```

```
        a = k-i+2; j = N/2^(a);
        For b = 1 to j do
            MVcA(a,b) = 2^(-0.5)(cA_{a-2,4b-2} +cA_{a-2,4b-1});
            MVcD(a,b) = 2^(-0.5)(cA_{a-2,4b-2}—cA_{a-2,4b-1});
        End for
End for
Construct TSTcA and TSTcD in terms of McA MVcA, McD, and MVcD;
Output TSTcA, and TSTcD;
Algorithm 2: TST-based single CP detection.
Input: two trigeminal search trees TSTcA and TSTcD built by Algorithm
1.
Initialization: C_1(α),C_2(β),C_3(γ), N = length(X^i), K = log_2 N, and
J = N/2^K, j = 1;
For i = 1 to K do // from the top root to the bottom leaf level;
    k = M-i+1;
    If (k >2) // dealing with nodes at the non-leaf level;
        Calculate variance fluctuations VF_{k,j:L}, VF_{k,j:R}, and VF_{k,j:M};
        VFmax = max(VF_{k,j:L}, VF_{k,j:R}, VF_{k,j:M});
        If (VFmax > C_1(α)) Then
          {Applying Criterion 2.1 to select the sub-branch with VFmax
          as the new current node, and add it into the resultant search
path;}
        else if (VFmax < = C_1(α)) Then
            Calculate statistic fluctuations SF_{k,j:L}, SF_{k,j:R}, and SF_{k,j:M};
            SFmax = max(SF_{k,j:L}, SF_{k,j:R}, SF_{k,j:M});
            If (SFmax > C_2(β)) Then
{Applying Criterion 2.2 to select the sub-branch with SFmax as the new
current node, and add it into the resultant search path;}
            else if (SFmax < = C_2(β)) Then
    {no abrupt change is detected and then cease the search procedure;}
                End if
            End if
    else if (k = = 1) // dealing with nodes in the bottom leaf level;
        Calculate statistic distances S'_L and S'_R;
        SDmax = max(S'_L, S'_R);
        If (SDmax > C_3(γ)) Then
{Applying Criterion 2.3 to select the leaf node with SDmax as the
resultant change point, and then add it into the search path;}
        else if (SDmax < = C_3(γ)) Then
            {no abrupt change is detected and then cease the search
procedure;}
            End if
    End if
End for
Output the resultant change point, and the resultant search path.
Algorithm 3: RSW&TST-based MCPs detection.
Input: An observed time series sample X' = {X_s, ..., X_e};
Initialization Cd, Cu, N_{X'} = length(X');
i = 1, Ws_i = We_i = 0, Nw_i = 0, TN_w_i = 0, N_R = N_{X'}, SetNw_i = {}, MCPs =
{};
while (N_R > = Cd) do
    if (N_R > = Cu)
        Tu_Nw_i = Cu;
    else
        Tu_Nw_i = N_R;
end If
Updating SetNw_i = {Nw_i| Cd < = Nw_i < = Cu};
```

```
Nw_i = Random(SetNw_i);  Ws_i = We_i +1;  We_i = Ws_i + Nw_i;
TN_w_i = TN_w_i + Nw_i;
N_R = N_X−TN_w_i;
ds = Ws_i;  de = We_i;
X^i = [X_1^i,...,X_Nwi^i] = [X_Ws_i,...,X_We_i];
CP_W_i = TST (X^i);  // Call TST-based single CP detection in Algorithm 2
MCPs = MCPs+{CP_Wi};// Adding CP_W_i into the resultant MCP vector
i = i+1;
end while
Output the resultant MCPs vector.
```

## Performance evaluation on MCPs detection

To evaluate the proposed RSW&TST performance for MCPs detection, the measurements including the hit, error, miss, and redundancy rates, as well as the search time are introduced. For a current slide window $W_i$ to be diagnosed, as shown in Fig 3, some related variables are defined in terms of the distance between the target CP (tCP) and the estimated CP (eCP) as follows:

1. Hit area: If a target CP named $tCP_i$ is located within a current slide window $W_i$, the hit area $HA_{tCPi}$ is formulated by $HA_{tCPi} = [tCP_i–hd_i, tCP_i+hd_i]$, where $hd_i$ is a distance constant.

2. Absolute distance: For an estimated change point called $eCP_i$ detected from $W_i$, the absolute distance $D_{tCPi}$ between the resultant $eCP_i$ and the actual $tCP_i$ is defined by $D_{tCPi} = |eCP_i–tCP_i|$.

3. Hit: Given an absolute distance $D_{tCPi}$ defined above, if $D_{tCPi} \leq hd_i$ holds, then the $tCP_i$ is considered to be hit by $eCP_i$, and marked by $Hit(tCP_i) = 1$.

4. Error: In contrast to a Hit, if $D_{tCPi} > hd_i$ holds, then $D_{tCPi}$ is dealt as an error of $tCP_i$, and recorded by $Error(tCP_i) = 1$.

5. Miss: For the target $tCP_i$ in the slide window $W_i$, if no change point is detected, then the $tCP_i$ is deemed as missed, and recorded by $Miss(tCP_i) = 1$.

6. Redundancy: If a resultant $eCP_i$ is falsely detected from $W_i$ when no target CP exists, then $eCP_i$ is identified as a redundancy, and recorded as $Redund(eCP_i) = 1$.

On the basis of these definitions above, the hit, miss, error, and redundancy rates, as well as search time, are then introduced as follows:

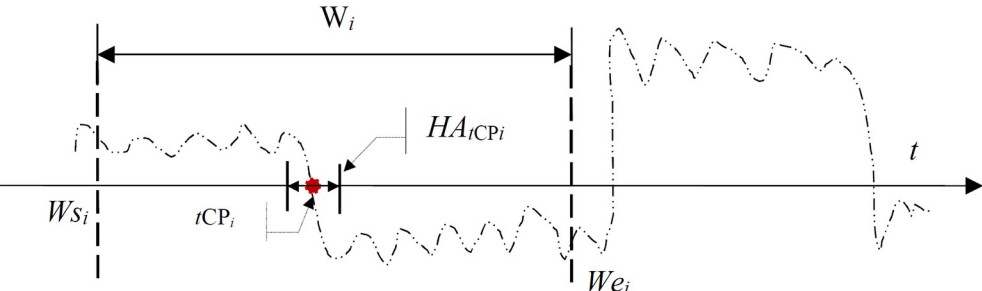

**Fig 3. The scheme of single CP detection within a slide window $W_i$ under the RSW approach.** The definitions of hit, error, miss and redundant are introduced according to the distance between the resultant $e$CP and the actual $t$CP respectively.

1. Hit rate: This is calculated as $\text{Hit}_{\text{rate}} = \left( N_{hit} / N_{tMCPs} \right) * 100\%$, where $N_{tMCPs}$ is the total number of the target MCPs named $t$MCPs, and $N_{hit} = \sum_{i=1}^{N_{tMCPs}} Hit(tCP_i)$, is the number of $t$MCPs that are hit by the estimated MCPs called $e$MCPs without error.

2. Miss rate: This is denoted as $\text{Miss}_{\text{rate}} = \left( N_{miss} / N_{tMCPs} \right) * 100\%$, where $N_{miss} = \sum_{i=1}^{N_{tMCPs}} Miss(tCP_i)$, is the number of $t$MCPs that are missed for all $e$MCPs.

3. Error rate: This is formulated as $\text{Error}_{\text{rate}} = \left( N_{error} / N_{tMCPs} \right) * 100\%$, where $N_{error} = \sum_{i=1}^{N_{tMCPs}} Error(tCP_i)$, is the number of $t$MCPs that exist error with $e$MCPs.

4. Redundancy rate: This is presented as $\text{Redun}_{\text{rate}} = \left( N_{redund} / N_{eMCPs} \right) * 100\%$, where $N_{eMCPs}$ is the total number of estimated MCPs, and $N_{redund} = \sum_{i=1}^{N_{eMCPs}} Redund(eCP_i)$, is the number of $e$MCPs that are redundant for all $t$MCPs.
   Obviously, it is true that $\text{Hit}_{\text{rate}}+\text{Miss}_{\text{rate+}}+\text{Error}_{\text{rate}} = 1$ for all of the target MCPs, and $\text{Redun}_{\text{rate}} = \left( 1 - \frac{N_{hit}}{N_{eMCPs}} \right) * 100\%$ holds for the total $e$MCPs.

5. Search time. This mainly consists of the TSTs construction and CP detection procedures, and is denoted by $ST_i = CT_i+DT_i$ for a slide window $W_i$. During the process of MCPs detection, search time is calculated as the average value of total search time in all slide windows, that is $ST = mean(\sum_{i=1}^{N_w} ST_i)$. Compared to some traditional algorithms of time complexity $O(N^2)$, such as KS, CUSUM or SSA, our TST-based method has a time complexity of about $O(\log N)$, and should theoretically be faster and more efficient.

## Results

In our synthetic simulation experiments, we evaluated the proposed RSW&TST framework by comparing it with the fixed slide window (FSW) approach and the CUSUM-based wild binary segment (WBS&CUS) method using metrics including hit, miss, error and redundancy rates respectively. Specifically, our TST method was verified against existing BST, KS and T methods under different RSW and FSW approaches. When our RSW&TST was applied for MCPs detection on different pathological signals in the clinical databases on PhysioNet [1,5,31], then the abnormal patterns were recognized in terms of the data features among resultant MCPs within abnormal data segments of epilepsy patients.

### MCPs detection on synthetic time series

In the synthetic experiments, a synthetic time series sample $X = \{X^1,\ldots,X^i,\ldots,X^N\}$ was assembled by $N$ pieces of data segments, in which $X^i = \{X_1^i, \ldots, X_j^i, \ldots, X_m^i\}$ was composed of the random numbers $N(\mu, \sigma)$ of size $m$, and the parameters $\mu$ and $\sigma$ were taken randomly from two sets $u = \{u_1, u_2,\ldots, u_N\}$ and $\sigma = \{\sigma_1, \sigma_2,\ldots, \sigma_N\}$ respectively. Therefore, the total $N$-1 target MCPs were assigned in the whole time series $X$.

In the first, a series of time series samples were synthesized with different numbers of target MCPs ranging from 30 to 120, then the proposed RSW&TST framework was tested by comparing it to BST, KS and T methods, respectively. As shown in Table 1, the results indicate that our TST has the highest hit-rate and the shortest computing time of all four methods, as well as relatively smaller miss, error and redundancy rates than most of the others. Specifically, the trend analyses in Fig 4 indicate that all tracks in the proposed TST keep more satisfactory

**Table 1. The mean analyses on MCPs detection under different numbers of *MCPs* from 2^6 to 2^15 in the RWS framework.**

| Methods | Time | Hit-rate | Miss-rate | Error-rate | Redund-rate |
|---------|------|----------|-----------|------------|-------------|
| RWS&TST | 0.0122 | 0.8930 | 0.1028 | 0.0095 | 0.0107 |
| RWS&BST | 0.0153 | 0.7662 | 0.1374 | 0.1089 | 0.1299 |
| RWS&KS | 0.6895 | 0.6146 | 0.0799 | 0.3478 | 0.3808 |
| RWS&T | 0.2831 | 0.7732 | 0.2268 | 0.0031 | 0.0040 |

levels, and more stable dynamics without drastic oscillations than the BST, KS and T methods in response to changeable MCPs, especially when the number of MCPs is much higher or lower.

The second simulations focus on testing the four methods above in the FSW framework, by using the sample of the fixed 30 *t*MCPs with different slide window sizes. Generally, the mean analyses listed in Table 2 reveal that our TST has relatively better performance due to having the shortest time and highest hit rate, as well as the lowest values of error and redundancy rates of all four methods. Unfortunately, all four methods have unsatisfactory and much lower efficiency than the former simulation results in the RSW approach. Furthermore, as for *Nw* ranging from 2^6 to 2^15, the trend analyses in Fig 5 show that our TST has much lower and more stable tracks of error and redundancy rates, but unstable hit and miss rates with more drastic fluctuations than the other three methods, especially when the size of *Nw* is much larger or smaller.

In addition, based on the synthetic sample with predefined 30 *t*MCPS under the TST-based FSW framework, some representative simulations were selected from the former experiments in Fig 5 above, then the results were plotted under *Nw* = 2^6, 2^11, and 2^15 respectively. The results shown in Table 3 and Fig 6 indicate that the TST-based FSW framework has the best efficiency as *Nw* takes a suitable value of 2^11, but much more sensitive and worse performance as *Nw* takes other values of 2^6, and 2^15. Therefore, it can be seen that a suitable size of slide window is very important for the efficiency of MCPs detection methods under the FSW framework.

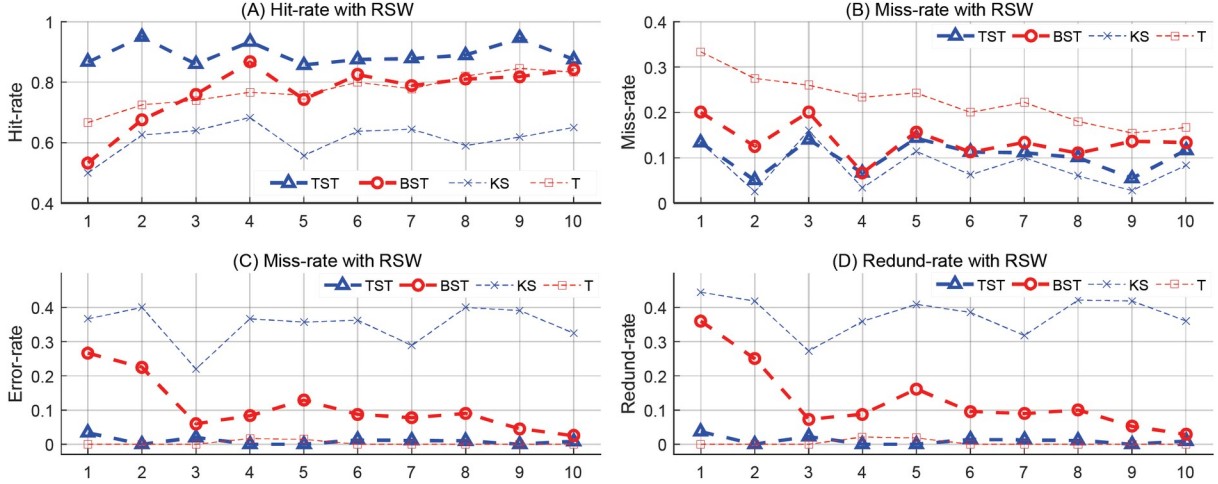

**Fig 4. The results of MCPs detection on a series of synthetic time series in the RSW approach.** For different numbers of MCPs from 30 to 120, the trend analyses on (A) Hit-rate, (B) Miss-rate, (C) Error-rate, and (D) Redund-rate, are illustrated by using TST, BST, KS and T methods respectively.

**Table 2. The mean analyses on MCPs detection under different sizes of *Nw* from 2^6 to 2^15 in the FWS framework.**

| Methods | Time | Hit-rate | Miss-rate | Error-rate | Redund-rate |
|---|---|---|---|---|---|
| FWS&TST | 0.0180 | 0.6633 | 0.3133 | 0.0033 | 0.0048 |
| FWS&BST | 0.0184 | 0.6233 | 0.2600 | 0.0100 | 0.1491 |
| FWS&KS | 0.5075 | 0.5867 | 0.2333 | 0.1933 | 0.3160 |
| FWS&T | 0.4562 | 0.5200 | 0.3767 | 0.1533 | 0.1527 |

In the final synthetic experiments, the simulations of MCPs detection under different numbers of *t*MCPs from 5 to 30 are implemented by using the WBS&CUSUM and the proposed RSW&TST respectively. Generally, the results in Fig 7 and Table 4 indicate that the WBS&CU-SUM tends to be unstable and inefficient as the number of *t*MCPs increases, due to smaller hit rate, bigger error and redundant rates, as well as longer search time. Especially, as shown in Fig 7 (F), a missed area appears in the last half of the time series sample, and none of the continuous *t*MCPs is detected from it. These results suggest that WBS&CUSUM is inefficient for MCPs detection on these synthetic datasets, probably because this 'greedy' method is hard to discern the transient and drastic data fluctuations from the regular and rhythmical oscillations, especially when the threshold of candidate change-point is unreasonable in the collection of CUSUM tests.

Meanwhile, the proposed RSW&TST is also executed for MCPs detection on these synthetic datasets. Compared with the WBS&CUSUM method above, as shown in Fig 8 and Table 5, the simulations indicate that our RSW&TST relatively keeps more stable and efficient as the number of *t*MCPs increases from 5 to 30, with higher hit rate, smaller missed, error and redundant rates, as well as shorter search time. These results suggest that our RSW&TST can successfully detect the target MCPs on these synthetic time series. A plausible reason is that it uses a global threshold of data fluctuation, and estimates the candidate change point orderly from the data segment in each of random slide windows, without any discarding or eliding operation in the MCPs detection procedure.

In summary, all these simulation results above verify that our RSW&TST framework has much better performance than that of both FSW approach and WBS&CUSUM method. It is an encouraging and efficient method for MCPs detection on large-scale time series.

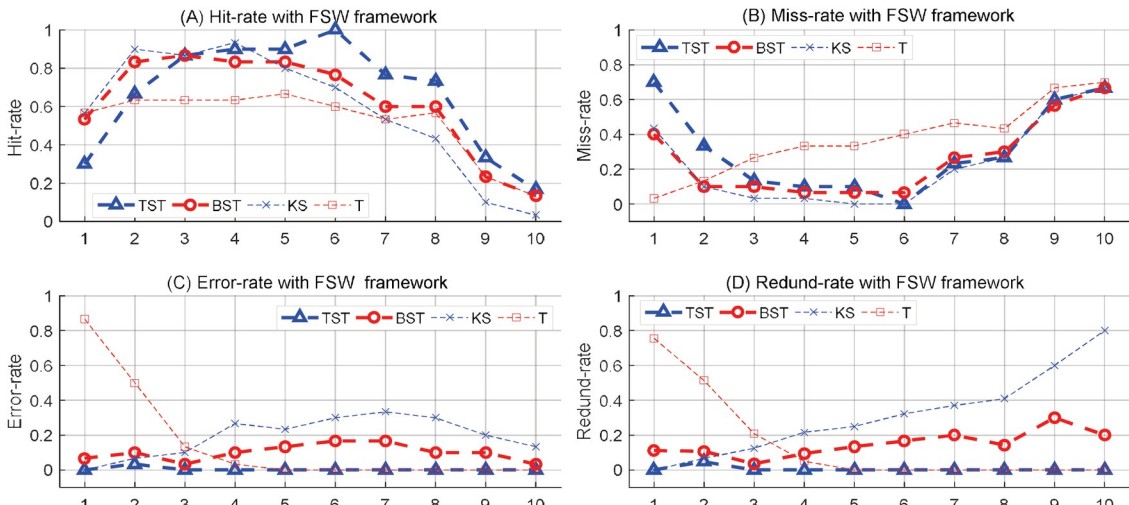

**Fig 5. The results of MCPs detection by using TST, BST, KS and T methods in the FSW framework.** For the synthetic sample with predefined 30 *t*MCPs, the trend analyses are illustrated in chart (A) Hit-rate, (B) Miss-rate, (C) Error-rate, and (D) Redund-rate, under different sizes of *Nw* from 2^6 to 2^15, respectively.

**Table 3. The analyses on MCPs detection as $Nw$ = 2^6, 2^11, and 2^15 in the TST-based FWS framework.**

| Value of $Nw$ | Hit | Miss | Error | Redund | Hit-rate | Miss-rate | Error-rate | Redun-rate |
|---|---|---|---|---|---|---|---|---|
| Nw = 2^6 | 9 | 21 | 0 | 0 | 0.300 | 0.700 | 0.000 | 0.000 |
| Nw = 2^11 | 30 | 0 | 0 | 0 | 1.000 | 0.000 | 0.000 | 0.000 |
| Nw = 2^15 | 5 | 25 | 0 | 0 | 0.167 | 0.833 | 0.000 | 0.000 |

### Abnormal pattern recognition on pathological signals

In the real data experiments, the proposed RSW&TST framework was used for MCPs detection on pathological recordings in the CAP sleep [5] and Post-Ictal Heart Rate Oscillations in Partial Epilepsy databases on PhysioNet [1,31]. First, it was evaluated by comparing it to the existing FWS and WBS&CUSUM approaches, as well as the BST, KS and T methods, respectively. Second, in terms of the numbers and positions of the resultant MCPs, the abnormal patterns were roughly recognized in the stage of a sudden epileptic attack. Last, the severity of the patients having the attack was roughly discerned based on the data features among MCPs within different abnormal areas.

In the first experiment, one ECG sample was selected from 22 pathological signals in the nfle10m, which is one of 40 recordings of patients diagnosed with nocturnal frontal lobe epilepsy (NFLE) in the CAP sleep databases [5]. Typically, the diagnosed ECG sample shown in Fig 9 can be roughly divided into two normal segments near the left and right parts, as well as an abnormal region within a sudden attack area called Sa1 in the middle. By means of the FSW framework, our TST was executed for MCPs detection on this ECG segment during a transient period of an epileptic attack. For different sizes of slide window $Nw$ from 2^10 to 2^14, the results in Fig 9(A)–9(E) show that the abnormal ECG region within Sa1 can be obviously discerned as $Nw$ = 2^12, in spite of a redundant CP Rc1 near the normal left part. However, as the size of $Nw$ is below the threshold of 2^12, the smaller $Nw$ is, the greater the number of redundant CPs; as a result, the Sa1 is harder to identify. Now, assume the size of $Nw$ is bigger than 2^12—the bigger $Nw$ is, the more missed CPs there are. Therefore, the Sa1 is harder to discern. These results above indicate that, as the size of $Nw$ takes a suitable threshold value, our TST method can roughly distinguish abnormal ECG segment area under a sudden epileptic

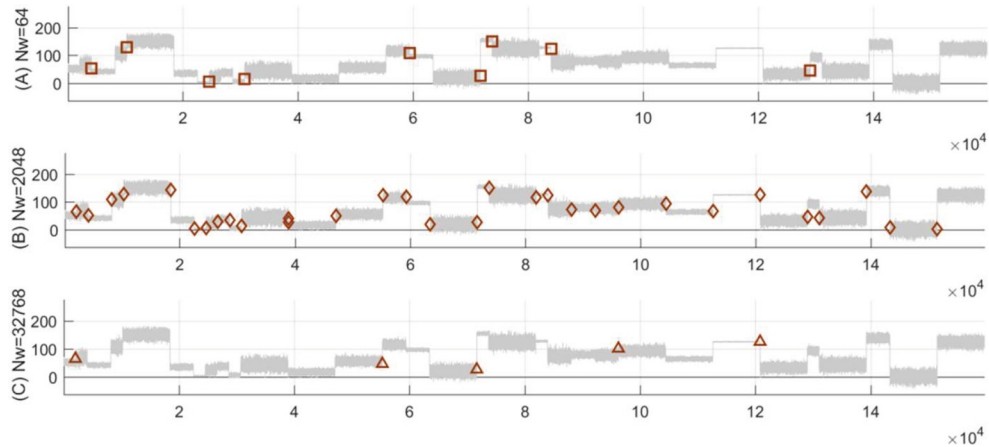

**Fig 6. The selected simulations of MCPs detection as $Nw$ taking a suitable, smaller or larger value in the FSW framework.** Given the similar sample with predefined 30 $t$MCPs used in Fig 5, the results of MCPs detection by using TST method are presented as (A) $Nw$ = 2^6, (B) $Nw$ = 2^11 and (C) $Nw$ = 2^15 respectively.

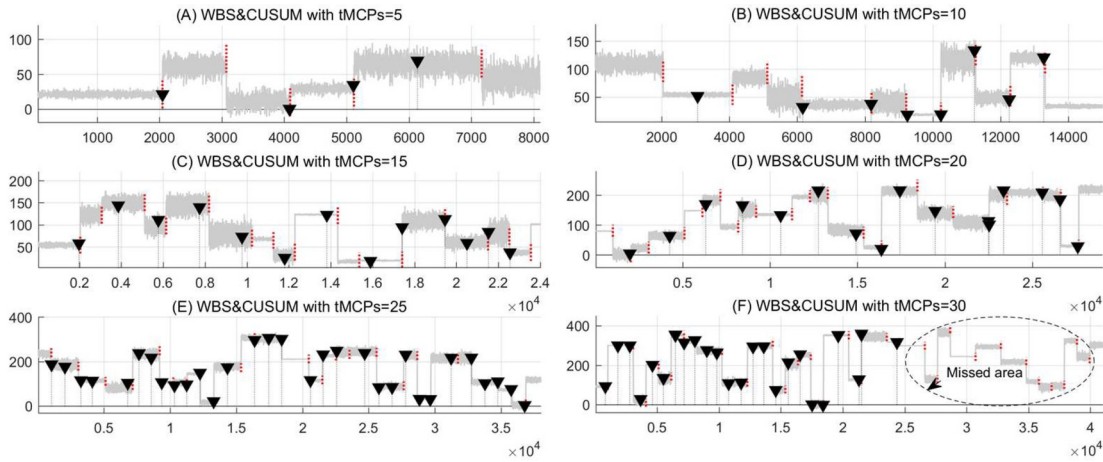

**Fig 7. The simulations of MCPs detection under different numbers of *t*MCPs in the WBS&CUSUM method.** For the synthetic time series with different *t*MCPs from 5 to 30, the results of MCPs detection are presented as (A) *t*MCPs = 5, (B) *t*MCPs = 10, (C) *t*MCPs = 15, (D) *t*MCPs = 20, (E) *t*MCPs = 25, and (F) *t*MCPs = 30, respectively.

attack, but it might be invalid as $Nw$ gets too big or too small. Unfortunately, it is very hard to determine an optimal threshold value of $Nw$ for the FSW framework, due to the complicated data features, especially for large-scale pathological bio-signals.

Meanwhile, our RSW&TST framework was tested further by using the same ECG sample with a sudden epileptic attack as above. Compared with existing BST, KS and T methods, the results shown in Fig 10 illustrate that our TSTKS can clearly distinguish the target abnormal segment area ASa1 between the resultant MCPs, without any redundant or missed points. However, for other three methods, the Asa1 is hard to discern according to the positions of resultant MCPs, due to the redundant points within the normal areas near the left and/or right parts, especially the missing beginning and/or ending in the target MCPs. These results suggest that the proposed RSW&TST framework can successfully distinguish the abnormal ECG segment under a sudden epilepsy attack, especially it does better without a suitable threshold of $Nw$ used in the FSW approach.

Moreover, the proposed RSW&TST and existing WBS&CUSUM were executed for MCPs detection on different ECG signals from sz04m.mat in the Partial Epilepsy databases [1,31]. The ECG segments were selected from the sz04m recording with different start points Dstart = 3800, 24800, 415000, and 949000, and then the resultant MCPs were detected by using our RSW&TST and existing WBS&CUSUM, respectively. Generally, as shown in Fig 11, in terms of different locations of the resultant MCPs, our RSW&TST can efficiently discern the abnormal areas with drastic fluctuations including AZ-A1, AZ-B1,B2,B3, AZ-C1, and

**Table 4. The results of MCPs detection as the *t*MCPs ranged from 5 to 30 in the WBS&CUSUM method.**

| Number of tMCPs | Hit | Miss | Error | Redund | Time | Hit-rate | Miss-rate | Error-rate | Redund-rate |
|---|---|---|---|---|---|---|---|---|---|
| *t*MCPs = 5 | 3 | 2 | 1 | 1 | .0032 | 0.600 | 0.400 | 0.200 | 0.250 |
| *t*MCPs = 10 | 7 | 2 | 1 | 1 | .0037 | 0.700 | 0.200 | 0.100 | 0.125 |
| *t*MCPs = 15 | 4 | 3 | 9 | 9 | .0040 | 0.267 | 0.200 | 0.600 | 0.692 |
| *t*MCPs = 20 | 5 | 6 | 9 | 10 | .0042 | 0.300 | 0.450 | 0.850 | 0.625 |
| *t*MCPs = 25 | 8 | 2 | 18 | 23 | .0044 | 0.320 | 0.080 | 0.720 | 0.741 |
| *t*MCPs = 30 | 3 | 13 | 15 | 21 | .0045 | 0.100 | 0.433 | 0.500 | 0.875 |

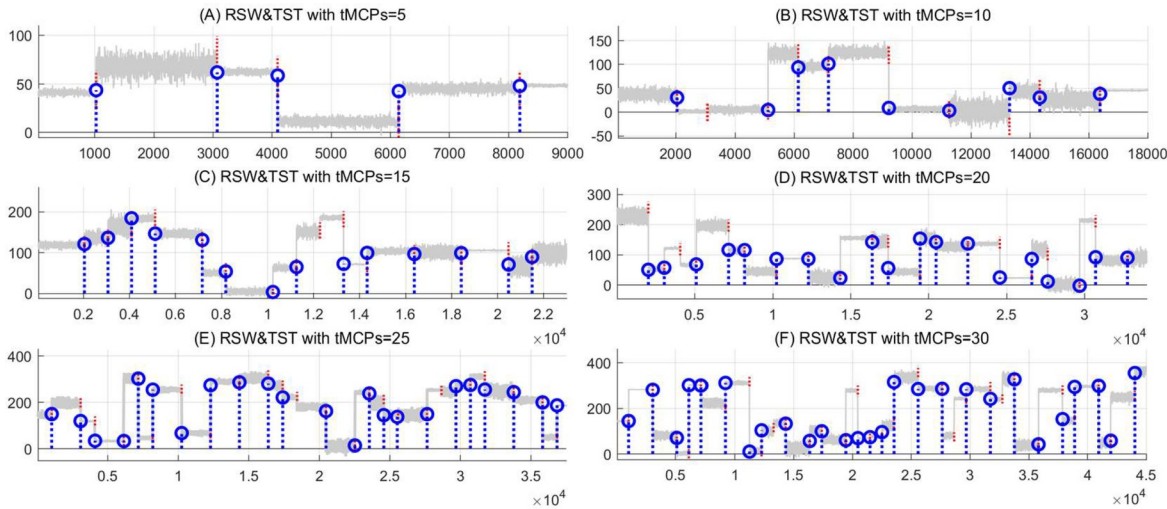

**Fig 8. The simulations of MCPs detection under different numbers of *t*MCPs in the RSW&TST framework.** For the synthetic time series with different *t*MCPs from 5 to 30, the results of MCPs detection are presented as (A) *t*MCPs = 5, (B) *t*MCPs = 10, (C) *t*MCPs = 15, (D) *t*MCPs = 20, (E) *t*MCPs = 25, and (F) *t*MCPs = 30, respectively.

AZ-D1, except of few redundant eCPs in Fig 11(A) and 11(D). However, the WBS&CUSUM method seems insensitive for the drastic fluctuations, especially some redundant eCPs are detected from the regular and rhythmic parts in these ECG signals. These results suggest that our RSW&TST is more efficient to recognize abnormal patterns from pathological ECG recordings of a patient in Post-Ictal Heart Rate Oscillations.

In the second experiment, our RSW&TST was executed for MPCs detection on pathological signals including ROC-LOC, SX1-SX2, EMG1-EMG2, Pleth, and Ox status, all of which were selected from the identical nfle10m of a patient diagnosed with NFLE in the CAP sleep data-bases [5], and then abnormal patterns were roughly recognized according to the numbers and positions of resultant MCPs obtained from different pathological signals. For the two EEG signals shown in Fig 12(A) and 12(B), the ROC-LOC seems more sensitive to a sudden NFLE attack, because it has the earliest start CP (EScp-a), as well as the continuous and dramatic fluctuations with bigger magnitude. On the other hand, the SX1-SX2 initially shows a slower response, and then has intermittent and intensive oscillations, as well as the latest end CP (LEcp-b). In addition, the EMG1-EMG2 in Fig 12(C) presents milder sensitivity due to weaker fluctuations and smaller swings, but has the largest number of resultant MCPs. For the Pleth in Fig 12(D), the periodical track is intermittently disrupted by irregular and moderate fluctuations. Fig 12(E) shows that the Ox status only takes several square waves with different widths, and has the latest start CP (LScp-e) and the earliest end CP (EEcp-e) of all five signals. To

**Table 5. The results of MCPs detection as the *t*MCPs ranged from 5 to 30 in the RSW&TST framework.**

| Number of tMCPs | Hit | Miss | Error | Redund | Time | Hit-rate | Miss-rate | Error-rate | Redund-rate |
|---|---|---|---|---|---|---|---|---|---|
| tMCPs = 5 | 5 | 0 | 0 | 0 | .0010 | 1.000 | 0.000 | 0.000 | 0. 000 |
| tMCPs = 10 | 9 | 1 | 0 | 0 | .0014 | 0.900 | 0.100 | 0.000 | 0.000 |
| tMCPs = 15 | 14 | 1 | 0 | 0 | .0015 | 0.933 | 0.067 | 0.000 | 0.000 |
| tMCPs = 20 | 19 | 1 | 0 | 0 | .0013 | 0.950 | 0.050 | 0.000 | 0.000 |
| tMCPs = 25 | 22 | 2 | 1 | 1 | .0013 | 0.880 | 0.080 | 0.040 | 0.043 |
| tMCPs = 30 | 27 | 3 | 0 | 0 | .0012 | 0.900 | 0.100 | 0.000 | 0.000 |

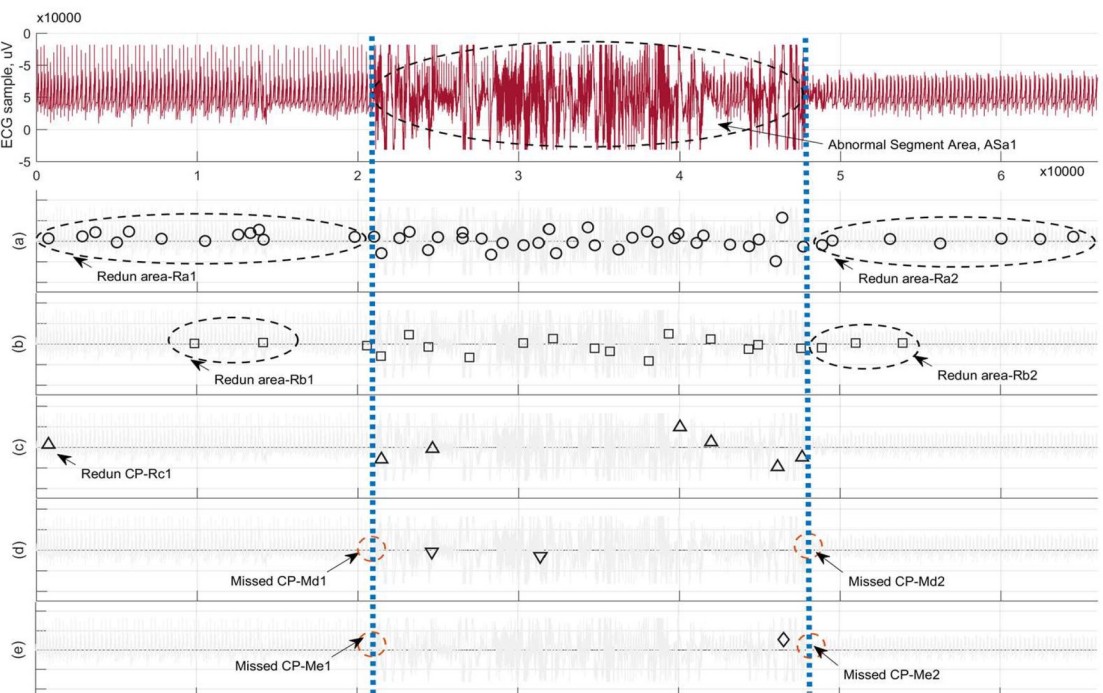

**Fig 9. The results of MCPs detection on ECG sample from one of 22 pathological signals in the nfle10_edfm by using TST method in the FSW framework.** In accordance with the partitions of ECG sample, the resultant MCPs are illustrated as the size of $Nw$ is equal to (a) 2^10, (b) 2^11, (c) 2^12, (d) 2^13, and (e) 2^14, respectively.

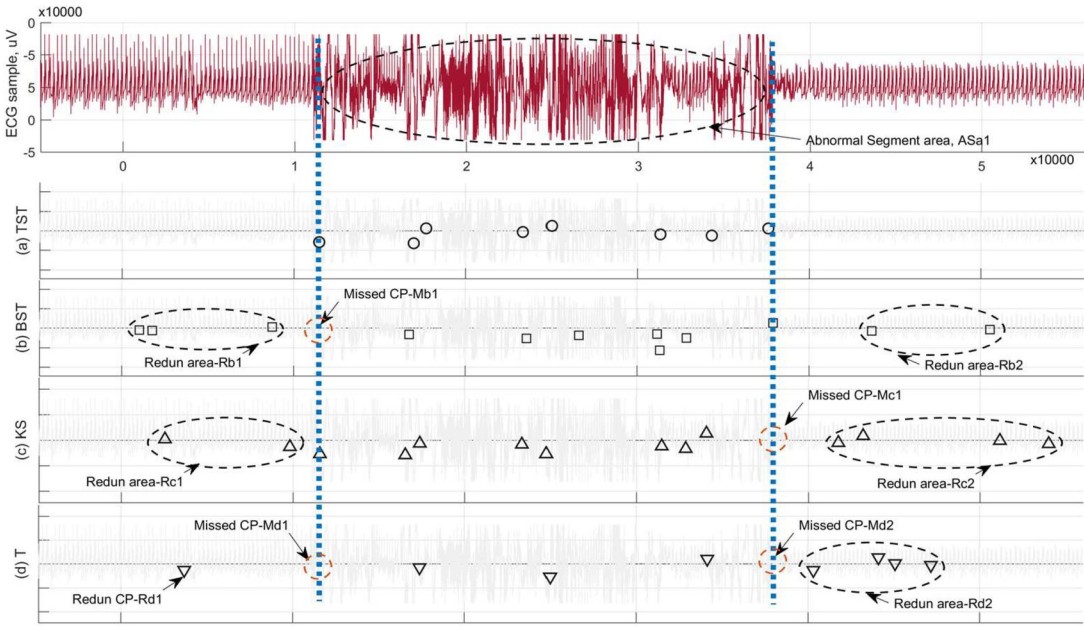

**Fig 10. The results of MCPs detection on the same ECG sample used in Fig 9 by means of different methods in the RSW approach.** In terms of the partitions of ECG sample, the resultant MCPs are illustrated by using (a) TST, (b) BST, (c) KS, and (d) T methods, respectively.

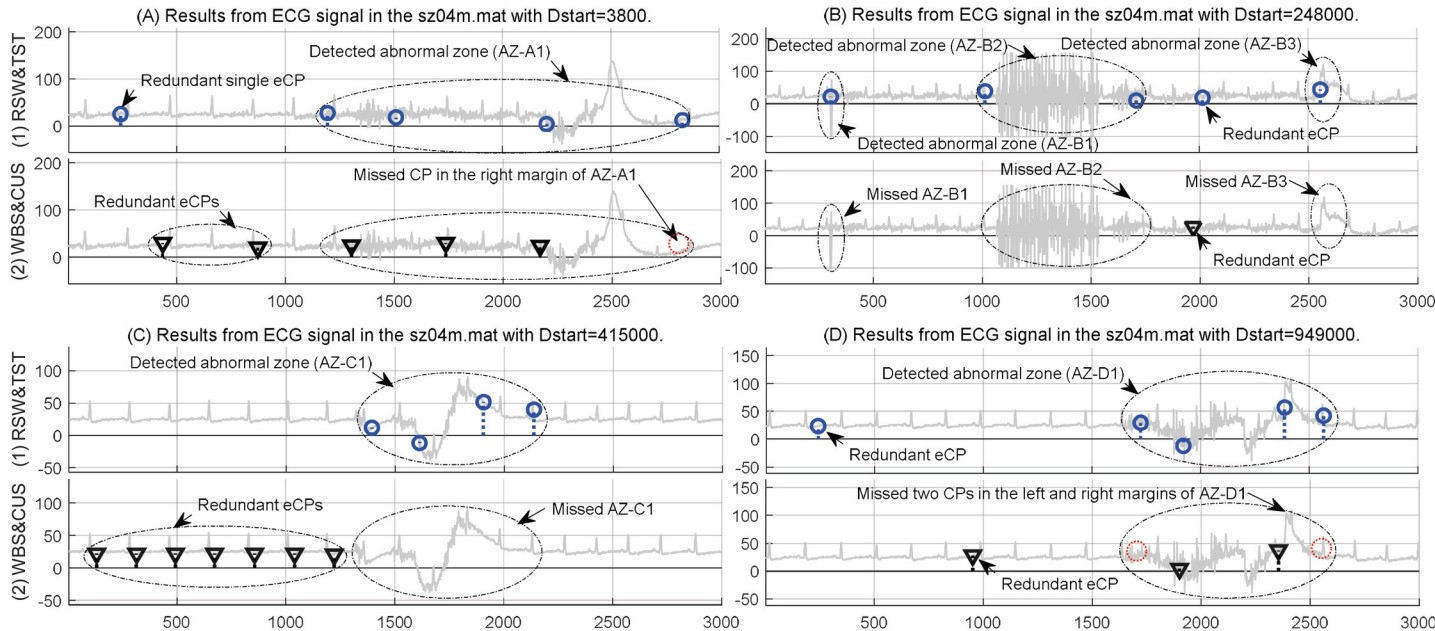

**Fig 11. The results of MCPs detection on different ECG segments from sz04m in the partial epilepsy databases.** By using the proposed RSW&TST framework and the WBS&CUSUM method, the resultant MCPs were detected from ECG signals in the sz04m.mat with the different start points of (A) Dstart = 3800, (B) Dstart = 24800, (C) Dstart = 415000, and (D) Dstart = 949000, respectively.

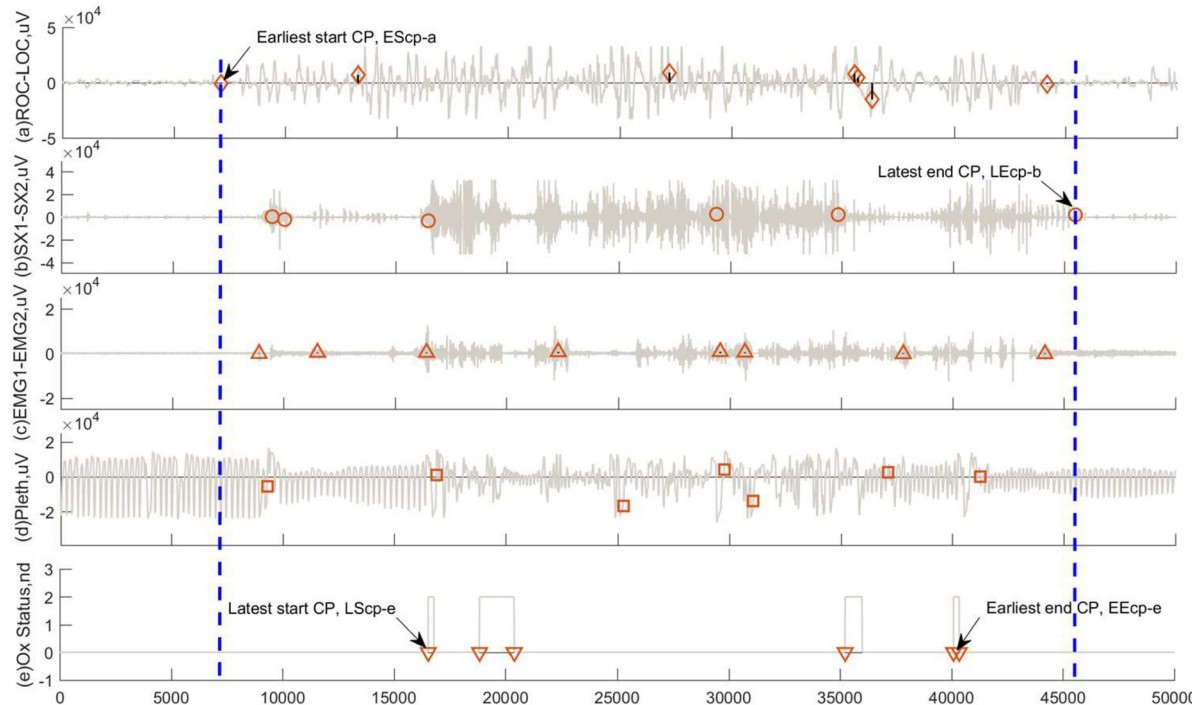

**Fig 12. The results of MCPs detection by using our RSW&TST framework on five pathological signals in the nfle10m recordings of a patient diagnosed with NFLE in the CAP sleep database.** During an identical period of NFLE sudden attack, the resultant MCPs are detected from two EEG signals (a) ROC-LOC, and (b) SX1-SX2, as well as other three signals (c) EMG1-EMG2, (d) Pleth, and (e) Ox Status, respectively.

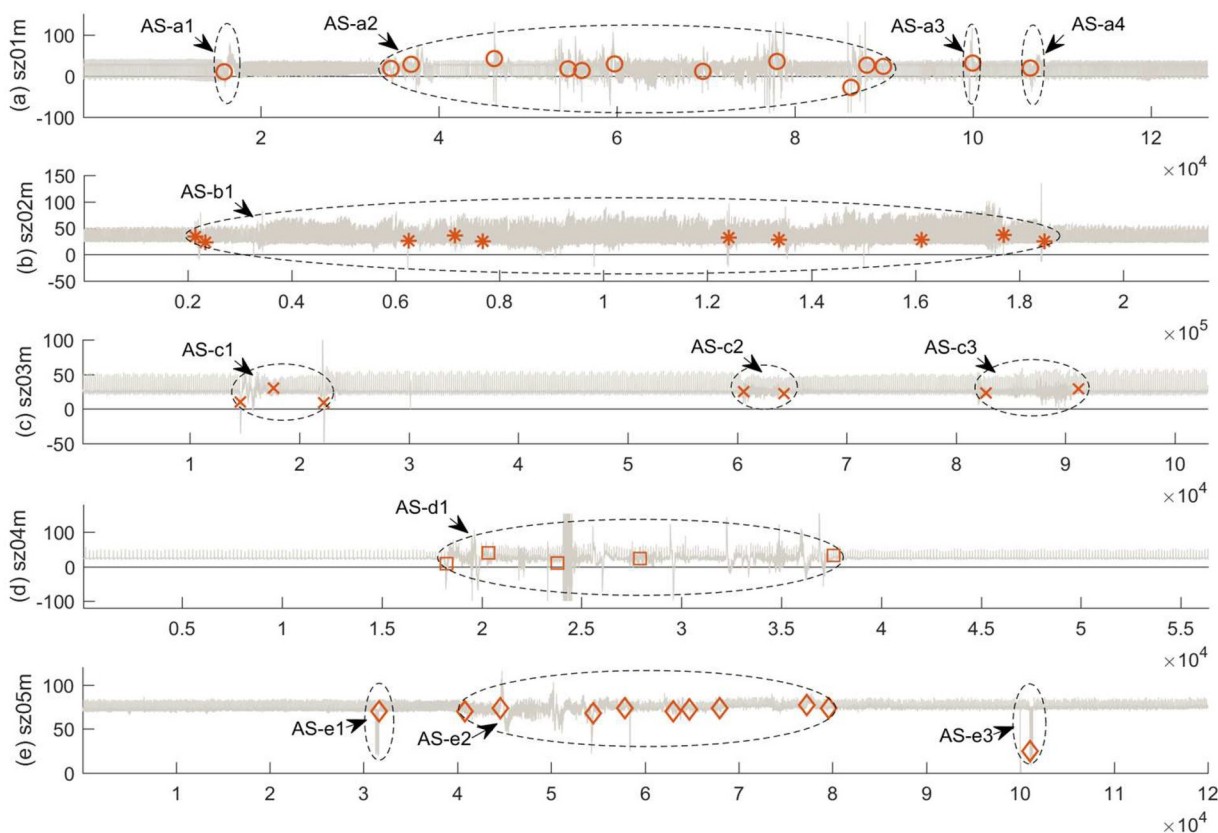

**Fig 13. The results of MCPs detection on five ECG signals from sz01m to sz05m in the partial epilepsy databases.** By using our RSW&TST framework, the resultant MCPs were detected on different ECG samples of patients in Post-Ictal Heart Rate Oscillations in the recordings of (A) sz01m, (B) sz02m, (C) sz03m, (D) sz04m, and (E) sz05m, respectively.

some extent, these experimental results above probably suggest that two EEG signals ROC-LOC and SX1-SX2 are most correlated with the NFLE attack, due to the data features of more drastic fluctuations among different MCPs during the process of NFLE attack. Therefore, EScp-a in ROC-LOC and LEcp-b in SX1-SX2 can roughly work as two indicators to predict the start and end during the period of a sudden NFLE attack.

In the last experiment, our RSW&TST was subsequently applied for MCPs detection on five ECG recordings of different patients in Post-Ictal Heart Rate Oscillations, which were selected respectively from sz01m to sz05m in Partial Epilepsy databases [1,31]. In our experiments, for the sz01m in Fig 13(A), it appears mainly as a bigger and more intensive abnormal segment AS-a2 with intermittent and sharp oscillations, which is composed of eleven gathered change points, and three smaller segments AS-a1, AS-a3 and AS-a4, which are locally scattered with a single change point. As for another sample sz02m in Fig 13(B), it roughly contains a whole abnormal area AS-b1 with ten change points in total, and it has more persistent and intensive fluctuations than those in the sz01m, and the longest lasting time of all five ECG samples. Compared with the sample sz02m, the sz04m in Fig 13(D) similarly has one abnormal segment AS-d1 including total five CPs, but it has shorter onset time, as well as more rapid and dramatic oscillations. For sz03m and sz05m in Fig 13(C) and 13(E) respectively, although both samples have the same three abnormal parts, sz05m with three abnormal areas is very similar to the sz01m except for having one more AS-a4, and sz03 looks more mild and has a slighter fluctuation in response to PE attack.

These experimental results suggest that abnormal segments in each of the ECG samples can be generally distinguished in accordance with the positions and numbers of the detected MCPs, and the severity of patients in the stage of partial epilepsy attack can be further evaluated in terms of the data features among different abnormal areas. Generally, the greater the number of MCPs is, the more abnormal zones there are. Specifically, the longer the time and the stronger the data fluctuations, the greater the severity of the partial epilepsy attack.

## Conclusions

In this paper, a novel RSW&TST framework was proposed for MCPs detection on large-scale time series. In our method, an observed data sample was first divided into a series of data segments by means of the random slide window strategy, in which the slide window size was stochastically chosen from a predefined collection in terms of data characteristics and experimental knowledge. Then, the piece of data segment in each slide window is diagnosed by using a TST-based CP detection procedure, and a potential change point is estimated from the top root to the bottom leaf nodes by using multi-channel search criteria in the target TST. Finally, the resultant MCPs were assembled by a series of single change points in each slide window.

In our synthetic simulations, our RSW&TST was evaluated by comparing it with the existing FSW and WBS&CUSUM, as well as BST, KS and SSA methods, in terms of computing time, the hit, missed, error and redundancy rates etc. The experimental results show that our RSW&TST has better performance because of a higher hit rate, as well as lower rates of computing time, miss, error and redundancy than other BST, KS and T methods in the RSW or FSW frameworks, as well as the WBS&CUSUM method, respectively. Furthermore, the proposed RSW&TST is applied for MCPs detection on pathological recordings of patients in partial epilepsy, and nocturnal frontal lobe epilepsy (NFLE) respectively. The experimental results of MCPs detection on different ECG signals also indicate that our RSW&TST has better performance than that of the existing FSW, WBS&CUSUM, and other BST, KS and T methods. Especailly, for each of the pathological signals, the abnormal parts are distinguished by the resultant MCPs, and the abnormal patterns are roughly recognized in terms of the numbers and positions of the resultant MCPs. Thus, the severity of patients in an epileptic state can be roughly analyzed based on the strength and duration of data fluctuations during the period of sudden epileptic attacks.

Our RSW&TST framework, although preliminary and simple, provides a novel and efficient method for MCPs detection quickly and efficiently, as well as a very flexible platform for abnormal pattern recognition from large-scale pathological signals.

## Acknowledgments

We would like to thank Prof.Qing Zhang and Prof.Mohan Karunanithi of the Australia e-Health Research Centre, CSIRO Computation Informatics, for their assistance, support and advice for this paper. Also, we appreciate the editors and referees for very helpful comments which led to a substantial improvement of this manuscript.

## Author Contributions

**Conceptualization:** Jinpeng Qi, Fang Pu.

**Data curation:** Jinpeng Qi, Fang Pu.

**Formal analysis:** Jinpeng Qi.

**Investigation:** Jinpeng Qi, Ping Zhang.

**Methodology:** Jinpeng Qi, Ying Zhu, Ping Zhang.

**Software:** Ying Zhu, Fang Pu.

**Supervision:** Ping Zhang.

**Validation:** Ying Zhu, Fang Pu, Ping Zhang.

**Writing – original draft:** Fang Pu.

**Writing – review & editing:** Fang Pu.

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
