## [Decision Letter · Decision Letter 0]

10 Sep 2021

PONE-D-21-21448A Novel RSW&TST Framework of MCPs Detection for Abnormal Pattern Recognition  on Large-scale time series and Pathological Signals in EpilepsyPLOS ONE

Dear Dr. Qi,

Thank you for submitting your manuscript to PLOS ONE. After careful consideration, we feel that it has merit but does not fully meet PLOS ONE’s publication criteria as it currently stands. Therefore, we invite you to submit a revised version of the manuscript that addresses the points raised during the review process.

We look forward to receiving your revised manuscript.

Kind regards,

Amita Nandal

Academic Editor

PLOS ONE

Journal Requirements:

"I would like to thank Prof. Mohan Karunanithi of the Australia e-Health Research Centre, CSIRO Computation Informatics, for his assistance, support and advice for this paper. This paper is supported by the National Natural Science Foundation of China (No.61104154), and the Specialized Research Fund of the Natural Science Foundation of Shanghai (no.16ZR1401300 and no.16ZR1401200)."

"This paper is supported by National Natural Science Foundation of China (No.61104154), and Specialized Research Fund for Natural Science Foundation of Shanghai (no.16ZR1401300 and no.16ZR1401200)"

Reviewers' comments:

Reviewer's Responses to Questions

**Comments to the Author**

1. Is the manuscript technically sound, and do the data support the conclusions?

Reviewer #1: Yes

Reviewer #2: Partly

2. Has the statistical analysis been performed appropriately and rigorously? 

Reviewer #1: Yes

Reviewer #2: Yes

3. Have the authors made all data underlying the findings in their manuscript fully available?

Reviewer #1: Yes

Reviewer #2: Yes

4. Is the manuscript presented in an intelligible fashion and written in standard English?

Reviewer #1: Yes

Reviewer #2: Yes

5. Review Comments to the Author

Reviewer #1: 1. State of the art technologies and algorithms achieve very good results.

2. Figure quality is not clear so try to improve it.

3. Proofread the manuscript for grammatical errors.

4. The readability, presentation and organization of this manuscript can be improved.

Reviewer #2: 1. The paper is not properly structured.

2. Proposed methodology section is not present. Instead the methods available in literature along with the proposed method is presented in Section 2.

3. It is strongly advised to restructure the paper by separating Background section and proposed method section.

4. Results section needs improvement. Include more comparison results with latest methods.

6. PLOS authors have the option to publish the peer review history of their article (what does this mean?). If published, this will include your full peer review and any attached files.

Reviewer #1: No

Reviewer #2: No

---

## [Author Response · Author response to Decision Letter 0]

17 Oct 2021

Reviewer #1: 1. State of the art technologies and algorithms achieve very good results.

2. Figure quality is not clear so try to improve it.

3. Proofread the manuscript for grammatical errors.

4. The readability, presentation and organization of this manuscript can be improved.

Response to Reviewer #1: 

(1) We have reedited and improved the quality of Figures.

(2) Including the corresponding author and all coauthors, we have adjusted the presentation and organization in the revised manuscript, 

(3) and then proofread the manuscript for grammatical errors, and improved the readability as we can do.

Reviewer #2: 1. The paper is not properly structured.

2. Proposed methodology section is not present. Instead the methods available in literature along with the proposed method is presented in Section 2.

3. It is strongly advised to restructure the paper by separating Background section and proposed method section.

4. Results section needs improvement. Include more comparison results with latest methods.

Response o Reviewer #2: 

According to the suggestions of reviewer, we have，

(1) restructured the paper by separating Background section and proposed method section in the revised version, 

(2) introduced another WBS&CUSUM method for MCPs detection, and added the related contents in the methodology part.

(3) and then, improved the result section by adding the comparison results of our RSW&TST with the WBS&CUSUM method, in the synthetic simulations and experiments on real pathological ECG datasets.

---

## [Editor Report · Decision Letter 1]

3 Nov 2021

A Novel RSW&TST Framework of MCPs Detection for Abnormal Pattern Recognition  on Large-scale time series and Pathological Signals in Epilepsy

PONE-D-21-21448R1

Dear Dr. Qi,

We’re pleased to inform you that your manuscript has been judged scientifically suitable for publication and will be formally accepted for publication once it meets all outstanding technical requirements.

Kind regards,

Amita Nandal

Academic Editor

PLOS ONE

Additional Editor Comments (optional):

I submit accept decision.
---

## [Editor Report · Acceptance letter]

3 Dec 2021

PONE-D-21-21448R1 

A Novel RSW&TST Framework of MCPs Detection for Abnormal Pattern Recognition  on Large-scale time series and Pathological Signals in Epilepsy 

Dear Dr. Qi:

I'm pleased to inform you that your manuscript has been deemed suitable for publication in PLOS ONE. Congratulations! Your manuscript is now with our production department. 

Kind regards, 

on behalf of

Dr. Amita Nandal 

Academic Editor

PLOS ONE